# Enhancement of Impact Abrasion Resistance Performances of White Cast Iron Utilizing Ti_3_AlC_2_

**DOI:** 10.3390/ma15165554

**Published:** 2022-08-12

**Authors:** Qi Dong, Haolin Li, Shuming Xing, Bo Qiu

**Affiliations:** 1College of Education, Zhejiang University of Technology, Hangzhou 310023, China; 2Department of Electrical Engineering, Tsinghua University, Beijing 100084, China; 3School of Materials Science and Engineering, Zhejiang Sci-Tech University, Hangzhou 310018, China; 4School of Mechanical and Electronic Control Engineering, Beijing Jiaotong University, Beijing 100080, China

**Keywords:** particle reinforcement, white cast iron, Ti_3_AlC_2_, TiC, casting, impact abrasion resistance

## Abstract

Al-Ti-C master alloy agent is currently the most promising grain refiner. This work investigates the influence of Ti_3_AlC_2_ addition (1.0–3.0 wt.%) on the microstructure of a hypoeutectic cast iron (4.7 wt.% Cr, 2.3 wt.% C). Microstructures of the samples were examined by SEM (scanning electron microscope). It was demonstrated that the added Ti_3_AlC_2_ did reduce the size of coarse primary carbides. The XRD (X-ray diffraction) pattern shows that Ti_3_AlC_2_ is decomposed into TiC in the alloy substrate. The EDS (energy dispersive spectrometer) interfacial element analysis shows that TiC combines well with the matrix interface. As the Ti_3_AlC_2_ amount was increased, the finest microstructure was achieved. When 2 wt.% Ti_3_AlC_2_ was added, the wear-resistance property of the material improved and became two times harder than the former. However, when 3% Ti_3_AlC_2_ was added, TiC gathered at the crystal boundary, which caused a decrease in the wear resistance of the material.

## 1. Introduction

The wear resistance of low Cr addition cast iron is worse than that of high-chromium cast iron with M_7_C_3_ carbides, but better than carbon steel and some low alloy steels [1,2,3,4]. It is necessary to reduce the brittleness of materials while maintaining wear resistance, and the contradiction between toughness and wear resistance needs to be dealt with [5,6,7].

The wide industrial application of chromium white cast irons has encouraged researchers to try different carbide-forming elements to improve the alloy microstructure, such as tungsten (W), vanadium (V), niobium (Nb), titanium (Ti) and boron (B), with the addition of a different carbide formation than cementite It reduces the carbon content of the matrix and improves the toughness and wear resistance of the material. Adding Ti in the cast iron to form TiC particles with a high melting point, as the heterogeneous nucleation substrate, and hinder the growth of carbides can refine the microstructure [8,9,10]. Therefore, with the increase in TiC content, the size of the carbides decreases, which is conducive to the improvement of hardness and toughness [11,12,13]. However, a large amount of TiC consumes the C element, which reduces the amount of carbide formation and hardness [14,15,16]. At the same time, the addition of C powder may change this phenomenon, but some of C powder floats into slag, which does not help the phenomenon of poor C [17,18]. Compared with directly adding TiC, the TiC formed in the matrix is finer, the size distribution in the metal matrix is more uniform, the microstructure of the matrix can be refined and it has good wettability with the black matrix, thus forming a stronger bonding interface [19,20,21].

Al-Ti-C master alloy agent is the most promising grain refiner at present. The low addition of Ti_3_AlC_2_ (about 1%) is mainly used as a refiner [22,23,24]. The thinning mechanism is that Ti_3_AlC_2_ is easy to decompose at a high temperature (above 870 °C) and the existence of a metal phase will promote the decomposition. The decomposed TiC particles, as a heterogeneous nucleation substrate, have better microstructure refinement effects to improve the mechanical properties [18,25,26,27]. Although alloy Al-Ti-C is refined by TiC particles, it is not as easily aggregated and subjected to Zr, Cr and TiB [28,29,30,31,32].

In this paper, the method of adding Ti_3_AlC_2_ in the melting and casting process of medium chromium alloy white iron is studied. The aim of this method is to refine the microstructure of cast iron, improve its wear resistance and achieve better wear resistance under the condition of low Cr addition. In addition, the interface of TiC generated by the decomposition of Ti_3_AlC_2_ in the process of alloy melting combines well with the metal matrix, and the strength formed is much better than its impact wear performance.

## 2. Experimental Method and Process

### 2.1. Experimental Materials

White cast iron. Raw cast iron is made by melting a certain proportion of steel, including scrap steel, ferromanganese, ferromolybdenum, etc. The chemical compositions of the alloys are presented in Table 1.

Ti_3_AlC_2_. A typical process follows: Ti, Al and C powder in Ti:Al:C = 3:1.1:2 mole ratio are mixed for 10 h and then shaped by cold pressing in a stainless steel mold to produce Ti_3_AlC_2_ powder. After 1 h pressureless sintering under 1450 °C, the sintered bulk sample shall be smashed into 10 μm (Figure 1).

### 2.2. Experimental Process

**Smelting.** In the experiment, 25 kg middle-frequency induction electric smelting in the laboratory is used, with melting temperature selected around 1400 °C, and fully melt raw materials are thermally insulated to preset pouring temperature to 1500 °C (infrared thermometer monitoring), and Ti_3_AlC_2_ powder should be added to the iron melt before pouring.

**Pouring.** Metal liquid shall be poured into a special concave and convex mold with a chamber size of 140 × 40 × 50 mm after discharge. Impact abrasive wear experiment is carried out on the MLD-10B abrasion machine, and the impact power is set as 1.5 J, impact frequency is set as 150 times/min, and rotating shaft speed is set as 200 r/min.

**Sampling.** The heat treatment of the sample will have an effect on the solidification structure and the secondary phase transformation will occur. Therefore, all specimens in this study were slowly cooled without heat treatment and were characterized and tested in the as-cast state. All specimens are taken from the same position on the samples. Metallographic sample and impact abrasive wear samples are, respectively, used for Rockwell hardness measurement and microstructure observation and impact abrasive wear experiment.

## 3. Results and Analysis

### 3.1. Microstructural Analysis

Under the addition of 1 wt.% Ti_3_AlC_2_, an abnormal eutectic carbide can be seen in Figure 2a, which is radially distributed in general and interlaced regularly with austenite. However, the carbides at the edge of the eutectic clusters are laterally connected to each other. The coarse eutectic carbides in Figure 2b are refined to a certain extent when the addition amount is 2.0 wt.%. The carbides are passivated at the sharp angle in Figure 2c, and the directivity disappears. On the contrary, the primary carbide is not refined, and the eutectic spacing becomes larger and larger with the increase in Ti_3_AlC_2_ content to 3 wt.% in Figure 2d.

It was found that the addition of 2 wt.% Ti_3_AlC_2_ was the most effective and weakened at 3 wt.%. As can be seen from the distribution of Ti elements in Figure 3, the Ti element is more dispersed when added in 1~2 wt.%. With the increase in the titanium content, the formation of TiC gradually increases and the average diameter gradually increases, as shown in Table 2. When the amount of TiC added is less than 2 wt.%, the average diameter is less than 1 μm. However, when the addition amount is more than 2.0 wt.%, the number of TiC particles increases rapidly and the diameter increases to more than 1 μm, and the distribution in the matrix is obviously uneven.

### 3.2. Effect of Ti_3_AlC_2_ on the Wear Resistance of the White Cast Iron

As seen in Figure 4, the hardness of the as-cast samples with different Ti_3_AlC_2_ addition amounts increases gradually with the increase in the addition amount, and the wear extent of the as-cast samples increases with increase in wear time. Among them, the non-addition sample has the biggest wear extent and worst wear-resistance property. The other three experimental conditions have relatively small wear extent, approximately increasing the wear property by 2 times compared to the non-addition sample. The sample with 2.0 wt.% Ti_3_AlC_2_ addition has the best wear performance. However, compared with 1.0 wt.% addition, wear loss under 3.0 wt.% addition increases, and is free from significant change after pressurization. This shows that the continuous addition of Ti_3_AlC_2_ does not contribute to the improvement of the wear-resistance property and may cause a negative effect.

As seen in Figure 5, the wear resistance increases with the increase in hardness. The sliding of impact abrasive particles makes the material to produce ploughs and spalling pits, which makes the metal break away from its parent body because the impact energy is 1.5 J. In Figure 5a, there are a number of cutting and chiseling marks on the grinding surface of metal mold casting, the cutting groove is deeper and the tearing takes away more matrix material. In Figure 5b, the grinding surface of the sample added with Ti_3_AlC_2_ is mainly ploughing and there is no spalling pit, which shows obvious abrasive wear and exists in the spalling pit, becoming very shallow. In Figure 5c, there are spalling pits at or near the end of the furrow, which indicates that the abrasive encountered hard carbides when it was pushed to these positions, and the carbides flaked off from the surface of the matrix after repeated impact and extrusion. Considering the same wear morphology shown in Figure 5a, deep spalling pits with large diameters are also observed in Figure 5d, which are identical to the size of primary carbides, indicating that larger areas of carbides are more prone to fracture, shedding and failure.

## 4. Discussion

### 4.1. Morphology and Distribution of TiC

By analyzing the microstructure and wear properties of white cast iron, the alloy with the Ti_3_AlC_2_ addition of 2 wt.% with uniform distribution are the best. In order to further explain the strengthening mechanism, magnification and elemental analysis were carried out on the joint site between particles and matrix, as shown in Figure 6. It can be seen that the smaller TiC becomes the nucleation basis of the matrix alloy after decomposition, which is also the reason for the refinement of carbide. The particles at lower temperatures, which do not dissolve completely and do not become nucleation centers, can also play a strengthening role. Both of the two types of particle boundaries have element diffusion with the matrix metal, forming reinforcement phase similar to the in situ formation, which is well combined with the interface between the matrix, and is not easy to cause the fall off of the reinforcement phase in the process of impact wear.

It is found that TiC aggregates at grain boundaries by adding 3.0 wt.% of the microstructures back scattering images and composition analysis as shown in Figure 7, which indicates that this may be due to the decomposition of a part of the TiC as the nucleation substrate to promote nucleation of primary austenite, while the other part of TiC, which is not the core, accumulates at the solidification front. Although the existence of these TiCs limits the liquid space of eutectic transformation, the liquid is divided into smaller regions and the growth space of eutectic carbides becomes narrower, which is beneficial to the refinement of the eutectic structure. However, the lattice distortion caused by it hinders the plastic deformation of metal materials at room temperature. During solidification, TiC is enriched at the front of the austenitic interface and carbide interface, respectively, to form a subcooled microzone, which increases the nucleation degree and hinders austenite and carbide growth. The effect of the above factors makes the occurrence of the crystallization process much easier.

Due to the uneven distribution of solute elements in alloy melts, the phase fraction ratio of solute in solid solution is different, which will lead to the difference of local component undercooling. Therefore, in the region with low undercooling, TiC particles struggle to grow. In the region with high undercooling degree, the grains with TiC particles as the nucleation will preferentially grow, and the latent heat will be released during the process of crystal formation. If the heat released by these preferential nuclei reduces the undercooling of the surrounding nuclei, then the TiC particles in the region with low undercooling will not form the nuclei and will eventually be enriched at the grain boundaries.

### 4.2. Strengthening Mechanism

From the point of view of particle theory, the AlTiC particle is difficult to melt and its structure is similar to the austenite. As shown in Figure 8, the lattice constant of austenite is a = 0.365 nm and the lattice constant of TiC = 0.432 nm; thus, the difference between them is very small. Austenite is easily nucleated directly on TiC according to the principle of structure and size adaptation. The eutectic transformation of the as-cast microstructure with the un-added Ti_3_AlC_2_ samples is almost carried out in the form of eutectic growth. When the first precipitated austenite is below the crystallization temperature, the austenite formed by eutectic will adhere to the eutectic growth. However, the existence of TiC hinders the possibility of the dependence growth and makes the structure transition to regular eutectic, which is shown in Figure 8. This is because the melting point difference between metal and non-metal eutectic is larger than that of solid phase, so the eutectic co-occurrence zone is inclined to the side with low melting point, which is more prominent. The two phases begin to grow together after entering the symbiosis zone.

## 5. Conclusions

(1) Ti_3_AlC_2_ addition can refine the eutectic carbide size of hypoeutectic white cast iron. With the increase in the Ti_3_AlC_2_ addition amount, the size of the carbide in the alloy is reduced. During the growth of the primary austenite, it is decomposed into TiC, which becomes the nucleation center of austenite. Because the lattice matching degree between austenite and TiC is higher, the growth of eutectic carbide is quicker than that of TiC, which slows down the growth rate of carbide and thus reduces the size.

(2) The dispersed TiC is prepared in situ by Ti_3_AlC_2_, and the element analysis shows that the boundaries of the TiC and matrix are tightly bound.

(3) Overall, 2.0 wt.%Ti_3_AlC_2_ addition provokes a two times improvement in the wear-resistance property of the material compared to the original one. Increasing the amount of Ti3AlC2 is helpful to improve the wear performance. However, excessive addition of Ti3AlC2 will make the generated TiC not become the core of inhomogeneous nucleation, but exist at the front of the solidification interface and aggregate at the grain boundary.

## Figures and Tables

**Figure 1 materials-15-05554-f001:**
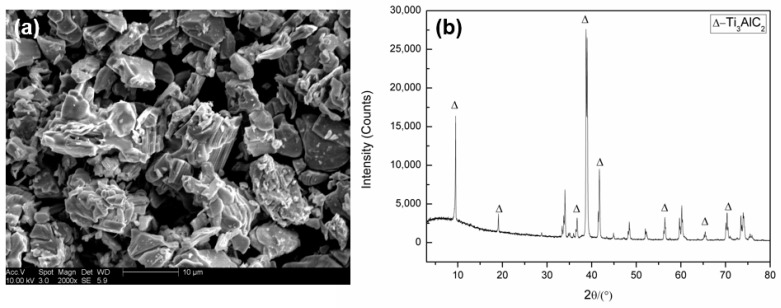
(**a**) Micromorphology and (**b**) composition of Ti3AlC2 powder.

**Figure 2 materials-15-05554-f002:**
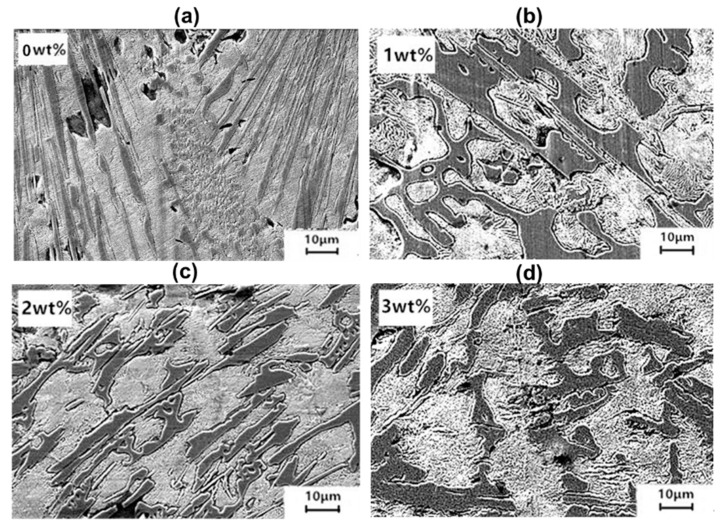
Microstructures of the white cast iron specimens with (**a**) 0 wt.% Ti_3_AlC_2_ additive; (**b**) 1 wt.% Ti_3_AlC_2_ additive; (**c**) 2 wt.% Ti_3_AlC_2_ additive; (**d**) 3 wt.% Ti_3_AlC_2_ additive.

**Figure 3 materials-15-05554-f003:**
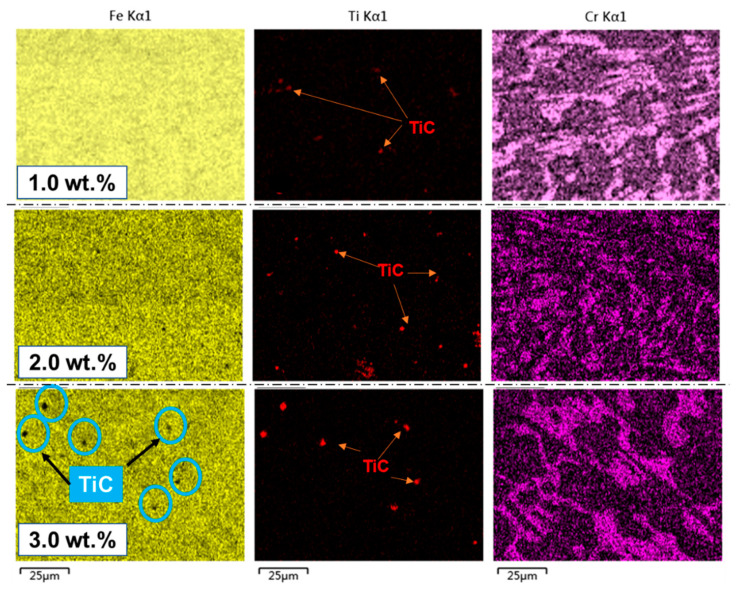
Distribution of TiC particle in the matrix of medium chromium cast iron with different Ti contents.

**Figure 4 materials-15-05554-f004:**
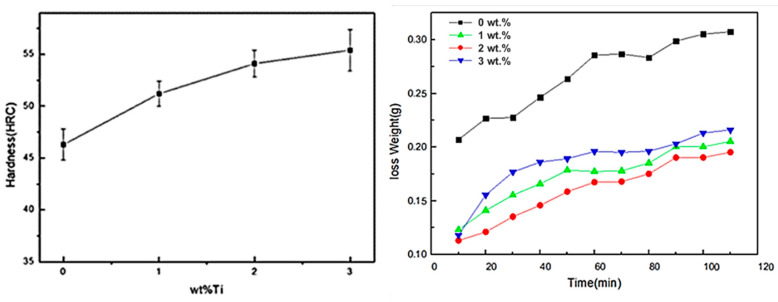
Hardness and wear curves of the white cast iron with different experimental conditions.

**Figure 5 materials-15-05554-f005:**
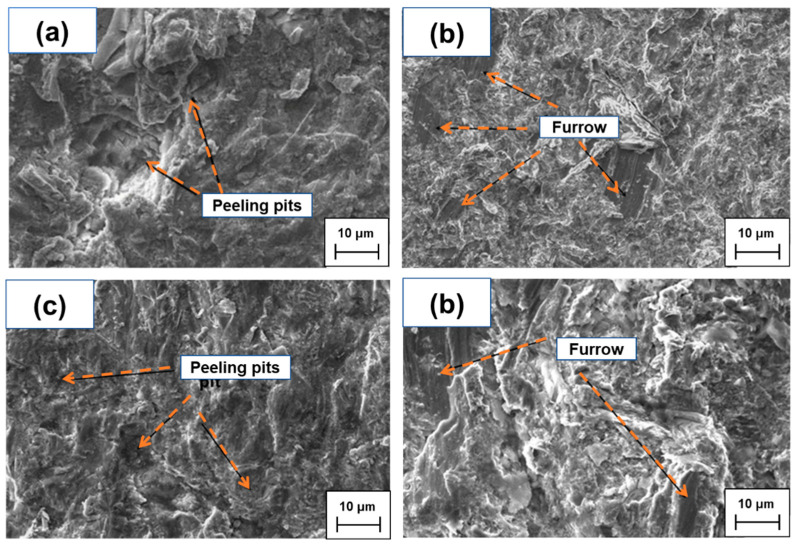
Wear surface with white cast iron (**a**) 0 wt. % Ti_3_AlC_2_ (**b**) 1.0 wt. % Ti_3_AlC_2_ additive. (**c**) 2.0 wt.% Ti_3_AlC_2_ additive (**d**) 3.0 wt.% Ti_3_AlC_2_ additive.

**Figure 6 materials-15-05554-f006:**
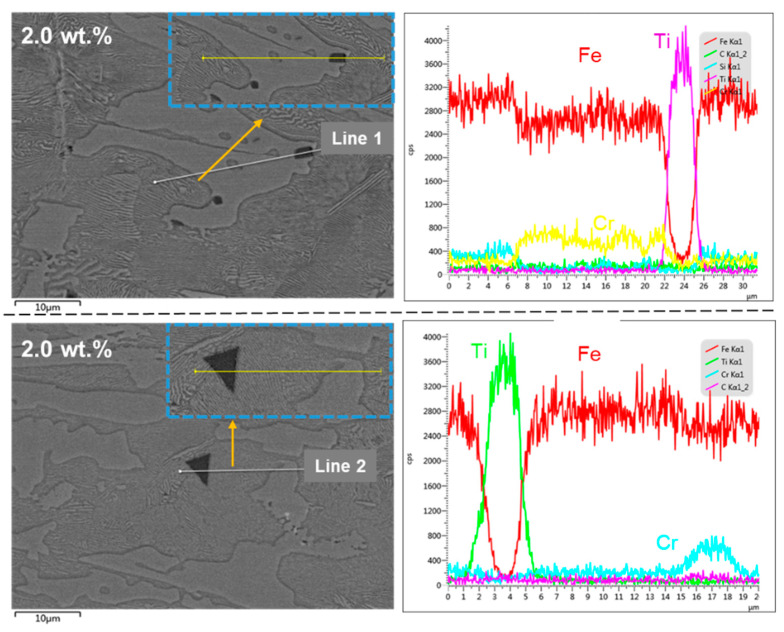
Distribution of TiC particle in the matrix of medium chromium cast iron with 2.0 wt.% Ti_3_AlC_2_ additive.

**Figure 7 materials-15-05554-f007:**
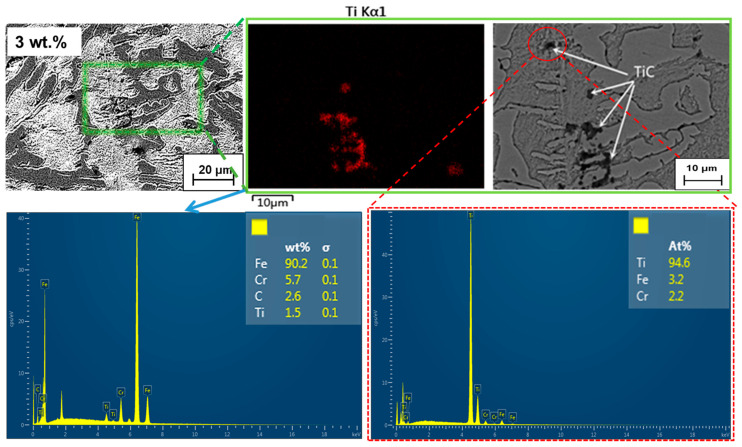
Representative backscatter images and EDS patterns of TiC in the samples under 3.0 wt.% Ti_3_AlC_2_. Schematic illustrations of TiC enrichment at the solid–liquid front.

**Figure 8 materials-15-05554-f008:**
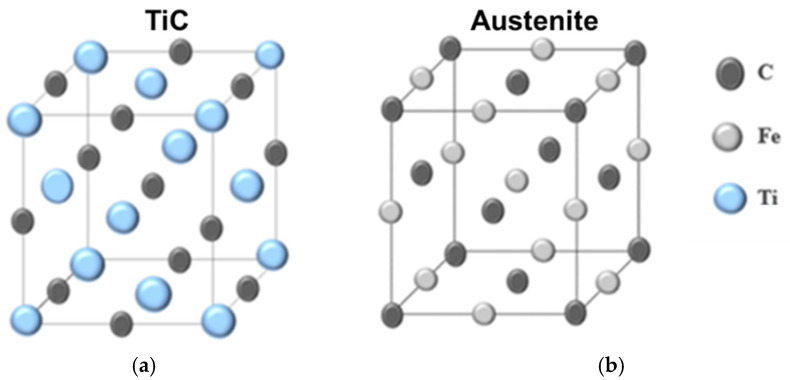
Schematic illustrations of (**a**) TiC and (**b**) austenite crystals.

**Table 1 materials-15-05554-t001:** Chemical compositions of the white cast iron.

Material	C	Cr	Si	Mn	Mo
ω/%	2.3	4.7	1.5	0.6	0.02

**Table 2 materials-15-05554-t002:** Fraction and average diameter of TiC with different Ti_3_AlC_2_ additives.

Content	Fraction (%)	Average Diameter (μm)
1.0 wt.%	1.91	0.85
2.0 wt.%	3.09	1.22
3.0 wt.%	3.57	1.71

## Data Availability

The datasets generated during and/or analyzed in the current study are available from the corresponding author on reasonable request.

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
