# Peer review of "Enhancement of Impact Abrasion Resistance Performances of White Cast Iron Utilizing Ti3AlC2"

_materials, 2022, doi:10.3390/ma15165554_

Round 1

Reviewer 1 Report

Review

Manuscript ID:  materials-1826110

Journal: Title: Materials

 Enhancement of impact abrasion resistance performances of 2 white cast iron utilizing Ti3AlC2

The author try to study   influences of Ti3AlC2 addition (1.0-3.0 wt.%) on microstructure of a hypoeu-9 tectic cast iron (4.7 wt.% Cr, 2.3 wt.% C). Microstructures of the samples were examined by SEM 10 (Scanning electron microscope). It was demonstrated that the added Ti3AlC2 did reduce the size of 11 coarse primary carbides. The XRD (X-ray diffraction) pattern shows that Ti3AlC2 is decomposed into 12 TiC in the alloy substrate. The EDS (Energy Dispersive Spectrometer) interfacial element analysis 13 shows that TiC is well combined with matrix interface. As the Ti3AlC2 amount was increased, the 14 finest microstructure was achieved. But there are some comments as the following:

1-    The author must add more keywords where not least than six words

2-    The author must add more references to introduction part

3-    The author must add more references to all parts of manuscript

4-    The author must update references because the present references are very old

5-    The author must add the novelty of this work

6-    the resolution must add more characterization such as XPS

7-    The author must modify fig.2 by add magnification power to compare

8-    The author must modify fig.4 the resolution is very bad

9-    The author must modify fig.6 the resolution is very bad

10-                     The author must modify fig.7 the resolution is very bad

Recommendation: Minor revision

Author Response

 The author must add more keywords where not least than six words

Our Response: Thanks for the reviewer’s comment. The keywords has been marked respectively (a) and (b), (c) and the image name was changed to surface morphology of Al with different thickness on PP film: (a) 20nm; (b) 40nm; (c) 100nm.

2-    The author must add more references to introduction part

3-    The author must add more references to all parts of manuscript

4-    The author must update references because the present references are very old

Our Response: Thanks for the reviewer’s comment. The format of references was modified according to the requirements. Some references have been added in the introduction section, and the date of the literatures has been updated to add literatures around 2020

5-    The author must add the novelty of this work

Our Response: Thanks for the reviewer’s comment. The novelty of this paper is that the strengthening effect of adding titanium carbide is higher than that of adding titanium element alone, and the volume ratio of titanium carbide formed is also higher.

6-    the resolution must add more characterization such as XPS

Our Response: Thanks very much for the reviewer s valuable suggestion. However, due to the covid-19 pendemic, many experimental facilities in our insititute are unavailable and the opening time is also uncertain. Hence, it is quite difficult for me to measure the corresponding XPS test at the moment.

7-    The author must modify fig.2 by add magnification power to compare

Our Response: Thanks very much for the reviewer s valuable suggestion. The magnification of the picture has been marked.

Figure 2. Microstructures of the white cast iron specimens with 500X :(a) 0 wt.% Ti3AlC2 additive; (b) 1 wt.% Ti3AlC2 additive; (c) 2wt.% Ti3AlC2 additive; (d) 3wt.% Ti3AlC2 additive.

8-    The author must modify fig.4 the resolution is very bad

9-    The author must modify fig.6 the resolution is very bad

10-   The author must modify fig.7 the resolution is very bad

Our Response: Thanks for the reviewer’s comment. The image(Fig.4\6\7) has been updated to a higher resolution.

Reviewer 2 Report

The paper is well written, the text is clear and easy to read.

The conclusions are in line with the evidence and arguments presented, but more detailed characteristics of the  wear curve are missing (significant slowing down of wear significant slowing down of wear between 60 a 80 min, 60 a 70 min respectively), the method of analysis of wear resistance is also not described.

HV hardness measurement over HRC may have been preferred and measured as well.

The results of the microstructure analysis are evaluated only qualitatively subjectively, the specific surface area of the particles could be measured (eg).

Other methods and their results are adequately used and described.

One can fully identify with the described results.

Author Response

Reviewer #1

The author try to study influences of Ti3AlC2 addition (1.0-3.0 wt.%) on microstructure of a hypoeu-9 tectic cast iron (4.7 wt.% Cr, 2.3 wt.% C). Microstructures of the samples were examined by SEM 10 (Scanning electron microscope). It was demonstrated that the added Ti3AlC2 did reduce the size of 11 coarse primary carbides. The XRD (X-ray diffraction) pattern shows that Ti3AlC2 is decomposed into 12 TiC in the alloy substrate. The EDS (Energy Dispersive Spectrometer) interfacial element analysis 13 shows that TiC is well combined with matrix interface. As the Ti3AlC2 amount was increased, the 14 finest microstructure was achieved. But there are some comments as the following:

1-    The author must add more keywords where not least than six words

Our Response: Thanks for the reviewer’s comment. The keywords has been marked respectively (a) and (b), (c) and the image name was changed to surface morphology of Al with different thickness on PP film: (a) 20nm; (b) 40nm; (c) 100nm.

2-    The author must add more references to introduction part

3-    The author must add more references to all parts of manuscript

4-    The author must update references because the present references are very old

Our Response: Thanks for the reviewer’s comment. The format of references was modified according to the requirements. Some references have been added in the introduction section, and the date of the literatures has been updated to add literatures around 2020

5-    The author must add the novelty of this work

Our Response: Thanks for the reviewer’s comment. The novelty of this paper is that the strengthening effect of adding titanium carbide is higher than that of adding titanium element alone, and the volume ratio of titanium carbide formed is also higher.

6-    the resolution must add more characterization such as XPS

Our Response: Thanks very much for the reviewer s valuable suggestion. However, due to the covid-19 pendemic, many experimental facilities in our insititute are unavailable and the opening time is also uncertain. Hence, it is quite difficult for me to measure the corresponding XPS test at the moment.

7-    The author must modify fig.2 by add magnification power to compare

Our Response: Thanks very much for the reviewer s valuable suggestion. The magnification of the picture has been marked.

Figure 2. Microstructures of the white cast iron specimens with 500X :(a) 0 wt.% Ti3AlC2 additive; (b) 1 wt.% Ti3AlC2 additive; (c) 2wt.% Ti3AlC2 additive; (d) 3wt.% Ti3AlC2 additive.

8-    The author must modify fig.4 the resolution is very bad

9-    The author must modify fig.6 the resolution is very bad

10-   The author must modify fig.7 the resolution is very bad

Our Response: Thanks for the reviewer’s comment. The image(Fig.4\6\7) has been updated to a higher resolution.

Reviewer #2

The paper is well written, the text is clear and easy to read.

The conclusions are in line with the evidence and arguments presented, but more detailed characteristics of the wear curve are missing (significant slowing down of wear significant slowing down of wear between 60 a 80 min, 60 a 70 min respectively), the method of analysis of wear resistance is also not described. HV hardness measurement over HRC may have been preferred and measured as well. The results of the microstructure analysis are evaluated only qualitatively subjectively, the specific surface area of the particles could be measured (eg).Other methods and their results are adequately used and described.One can fully identify with the described results.

Our Response: Thanks very much for the reviewer s valuable suggestion. However, due to the covid-19 pendemic, many experimental facilities in our insititute are unavailable and the opening time is also uncertain. Hence, it is quite difficult for me to measure the corresponding test at the moment.
